# Protein Interactomic Analysis of SAPKs and ABA-Inducible bZIPs Revealed Key Roles of SAPK10 in Rice Flowering

**DOI:** 10.3390/ijms20061427

**Published:** 2019-03-21

**Authors:** Xixi Liu, Zhiyong Li, Yuxuan Hou, Yifeng Wang, Huimei Wang, Xiaohong Tong, Hejun Ao, Jian Zhang

**Affiliations:** 1State Key Lab of Rice Biology, China National Rice Research Institute, Hangzhou 311400, China; 18338690086@163.com (X.L.); lzhy1418@163.com (Z.L.); houyuxuan@caas.cn (Y.H.); wangyifeng@caas.cn (Y.W.); wangyingkai2006@126.com (H.W.); tongxiaohong@caas.cn (X.T.); 2College of Agricultural Sciences, Hunan Agricultural University, Changsha 410128, China

**Keywords:** ABA, flowering time, SnRK2s, bZIP, SAPK, rice (*Oryza sativa* L.)

## Abstract

As core components of ABA signaling pathway, SnRK2s (Sucrose nonfermenting1–Related protein Kinase 2) bind to and phosphorylate AREB/ABF (ABA responsive element binding protein/ABRE-binding factor) transcriptional factors, particularly bZIPs (basic region-leucine zipper), to participate in various biological processes, including flowering. Rice contains 10 SnRK2 members denoted as SAPK1-10 (Stress-Activated Protein Kinase) and dozens of bZIPs. However, which of the SAPKs and bZIPs pair and involve in ABA signaling remains largely unknown. In this study, we carried out a systematical protein-protein interactomic analysis of 10 SAPKs and 9 ABA-inducible bZIPs using yeast-two-hybrid technique, and identified 14 positive interactions. The reliability of Y2H work was verified by in vitro pull-down assay of the key flowering regulator bZIP77 with SAPK9 and SAPK10, respectively. Moreover, SAPK10 could phosphorylate bZIP77 in vitro. Over-expression of SAPK10 resulted in earlier flowering time, at least partially through regulating the FAC-MADS15 pathway. Conclusively, our results provided an overall view of the SAPK-bZIP interactions, and shed novel lights on the mechanisms of ABA-regulated rice flowering.

## 1. Introduction

Abscisic acid (ABA) serves as an endogenous messenger that modulates various biological processes such as seed dormancy and germination, floral transition, seedling growth, biotic and abiotic stress responses [1]. Based on the knowledge from Arabidopsis, ABA signaling is majorly transmitted through a “PYR/PYL/RCAR-PP2C-SnRK2-AREB/ABF” cascade [2,3,4]. In this model, signals are perceived by ABA receptor complex PYR (PYrabactin Resistance 1) /PYL (PYR1-like) /RCAR (Regulatory Components of ABA Receptor)-PP2C (2C-type protein phosphatase), and transmitted to SnRK2s. Activated SnRK2s further pass the signals to AREB/ABF TFs in the post-translational level, majorly through protein phosphorylations on their conserved motifs like RXXS/T (where X stands for any amino acids, except S, T or Y) [5], while AREB/ABF (ABRE-binding protein/ABRE-binding factor) transcription factors (TFs) finally activate the expression of down-stream genes to respond to the growth signals or environmental cues. Given the essential roles of SnRKs-AREB/ABF regulatory pathway in ABA signaling, identification of SnRK2s and AREB/ABF TFs involved in ABA signaling has been a hot topic in the past decades. In Arabidopsis, SRK2D/SnRK2.2, SRK2E/SnRK2.6 and SRK2I/SnRK2.3 are crucial for ABA signaling, as the triple mutants almost completely lost the sensitivity to ABA [4]. Numerous AREB/ABF TFs have also been identified, of which the majority are bZIP (basic region-leucine zipper) TFs [5,6]. A few examples are ABI5 and AtDPBF2 which are implicated in seed maturation and germination, and ABF1, ABF2 and ABF4 which are related to abiotic stress responses [7,8,9,10]. SnRK2-mediated phophorylation on ABI5 could delay flowering by promoting the flowering suppressor FLC (FLOWERING LOCUS) expression [11].

Based on the results from genome survey, rice SnRK2 protein family contains 10 members denoted as SAPK1-10 (osmotic Stress/ABA-activated Protein Kinase) [12]. It was demonstrated that only SAPK6, SAPK8, SAPK9 and SAPK10 showed ABA-inducible patterns, and are functionally related to hyper osmotic stress signaling [13,14]. SAPK10 is able to phosphorylate TRAB1/bZIP66 on Ser94 and Ser102 [13]. It was also evidenced that SAPK6 interacted with and phosphorylated bZIP10 in response to ABA signaling in rice [14], while OsbZIP46 was phosphorylated and activated by SAPK6 and SAPK9 responding to ABA signaling and drought stress tolerance in rice [15,16]. However, SAPK2 recently was also found to be able to phosphorylate OsbZIP23 and bZIP46 for their transcriptional activation in drought resistance, which further expanded our understanding on the function of SAPKs in ABA signaling [15,16,17]. Despite the gained knowledge on this field, the identification of SAPK-regulated AREB/ABF TFs is rather poor in rice, considering a wide range of rice TFs, particularly bZIPs, that participate in the ABA signaling. Assuming that rice SAPKs have conserved functions to bind and phosphorylate AREB/ABF TFs, just like their orthologs in Arabidopsis, it will be of great interests to systematically identify SAPK-interactive bZIPs, and test their potential roles in ABA signaling. Here, we report a systematical, interactomic analysis of 10 rice SAPKs and 9 ABA-inducible bZIPs using yeast-two-hybrid (Y2H) technique. We finally identified 14 positive interactions between SAPKs and 9 bZIPs, and further verified bZIP77/OsFD1 interaction with SAPK10 and SAPK9 in vitro by Pull-Down assay. Moreover, SAPK10 could phosphorylate bZIP77/OsFD1 in vitro. In accordance to the positive roles of bZIP77 in rice flowering, over-expression of SAPK10 promotes heading under SDs and LDs. Our research provides new insights into the connections of SAPKs and ABA-inducible bZIPs, as well as the role of SAPK10 in rice flowering time regulation.

## 2. Results

### 2.1. Identification of SAPKs and ABA-inducible bZIPs

According to the public tissue expression pattern database (Rice Gene Expression; http://signal.salk.edu/cgi-bin/RiceGE5), nine *bZIP* transcription factors were drastically induced by ABA treatment, including *bZIP10*, *bZIP11*, *bZIP35*, *bZIP52*, *bZIP55*, *bZIP71*, *bZIP75*, *bZIP77* and *bZIP83*. Some ABA-inducible *bZIPs*, such as *bZIP23*, *bZIP46* and *TRAB1/bZIP66* were not included in the assay, as their protein-protein interaction relationship with all the SAPKs have been elucidated in previous studies [12,13,17]. To confirm the ABA-responsive transcriptional pattern of these genes, we performed qRT-PCR (quantitative Real Time-Polymerase Chain Reaction) to examine the transcriptional levels of young seedlings at a serial of time points after ABA treatment. All the tested *bZIPs* were significantly elevated at 12 HAT (Hours After Treatment), and the up-regulated level ranged from 10 to 1400 folds. In addition, we also checked the transcriptional response of *SAPK1-10* to ABA treatment. Robust transcriptional induction was only found in *SAPK9*, which showed over 20 folds up-regulation at 8 HAT, while the other genes only showed moderate changes of gene expression, which is consistent to the previous studies [12] (Figure 1).

### 2.2. Interactomic Analysis of SAPKs and ABA-inducible bZIPs in Yeast

SnRK2 kinases physically bind to and phosphorylate ABFs, particularly bZIPs, to transmit ABA signals in plant [2,18]. However, what and how rice bZIPs are involved in ABA signaling remains largely unknown. In the current study, we systematically screened the protein-protein interactions between SAPKs and ABA-inducible bZIPs to address the questions above. Owing to its simplicity and high through-put, Y2H has been widely used for the detection of physical protein-protein interactions [19]. Employing the Y2H system, we screened a total of 110 combinations (9 bZIPs plus empty pDEST32 as baits; 10 SAPKs plus empty pDEST22 as preys). Except that bZIP52 showed self-activation in the Y2H system, all the remaining tested bZIPs found at least one interactive SAPK (Figure 2 and Table 1). Among the 14 positive interactions, bZIP35 and bZIP77 had three interactive SAPKs, while the other tested bZIPs had only one or two interactive SAPKs. In terms of SAPKs, we also found SAPK9 and SAPK10 had more interactive bZIPs than other family members.

### 2.3. SAPK10 Phosphorylates OsbZIP77

To verify the interactions identified in Y2H, we did in vitro GST pull-down assay for the combinations of SAPK9-bZIP77 and SAPK10-bZIP77. In the assay, SAPK9 and SAPK10 were fused with GST tag, while bZIP77 was fused with HIS tag. The recombinant proteins together with pure GST tag were purified from *E. coli* and applied for pull down using Glutathione High Capacity magnetic Agarose Beads. The result showed that both SAPK9 and SAPK10 were directly co-immunoprecipitated with OsbZIP77, whereas the GST tag alone was not, indicating the high reliability of our interactomic results (Figure 3A,B). Given the kinase feature of SAPKs, we further did kinase assay to determine the kinase-substrate relationship between them and bZIP77. As shown in Figure 3C and Appendix A, HIS-OsbZIP77 was phosphorylated by GST-SAPK10, but not GST-SAPK9 and GST tag. To confirm the detection of phostag is specific on phosphorylated protein, we treated the protein sample with CIAP, which removes phosphate groups on proteins [20,21]. The phosphorylated band disappeared after CIAP treatment, suggesting that the phosphorylation is highly specific. In a previous publication, bZIP77 has been reported as OsFD1, and could form FAC (florigen activation complex) with 14-3-3 and Hd3a heterodimer in the nuclear to promote flowering in rice. Knock-down of *OsFD1* by RNAi resulted in later flowering, while over-expression of it could lead to earlier flowering [22]. More interestingly, it was demonstrated that phosphorylation on 192^nd^ serine of OsFD1 is indispensable in the formation of functional FAC in nucleus [22]. Nevertheless, it remains unknown that which kinases conduct the phosphorylation on OsFD1. Though we have revealed that SAPK10 is one of the candidate kinases of OsFD1, specifying out the SAPK10-mediated phosphorylation on the 192^nd^ serine of FD1 will be of great importance in confirming the function of SAPK10 in FAC formation and flowering regulation.

### 2.4. Over-Expression of SAPK10 Confers early Flowering

As core element in rice ABA signaling, SAPK10 has been implicated in seed germination, post-germination growth and root hair development [23,24]. However, the potential functions of SAPK10 in rice flowering remain unclear. Given the key roles of bZIP77 in rice flowering, we speculated that SAPK10, as the upstream kinase, may also be involved in flowering regulation. We therefore generated SAPK10 RNAi and over-expression lines, and investigated their heading date under both LD (Long day) and SD (Short day) conditions. The SAPK10 RNAi lines showed severe reduction of the gene transcription (Appendix A). However, no difference in flowering date were observed when compared with the WT, possibly due to the functional redundancy with other SAPK members such as SAPK9 and SAPK8 (Appendix A). Two representative lines OxSAPK10-1 and OxSAPK10-2 were chosen for the experiment due to their highly over-expressed levels when compared with the WT (Figure 4D). Similar to the early flowering observed on *OsFD1* over-expression lines, the OxSAPK10s also displayed around 14 days and 7 days of earlier heading than the WT in LD and SD conditions, respectively (Figure 4A–C). Subsequently, we examined the transcriptional levels of *bZIP77* and its target gene *MADS15* in OxSAPK10 lines. The expressions of both *bZIP77* and *MADS15* were significantly elevated in OxSAPK10-1 and OxSAPK10-2, indicating SAPK10 regulates rice flowering, at least partially, through the FAC-MADS15 pathway (Figure 4E,F).

## 3. Materials and Methods

### 3.1. Plant Materials and ABA Treatment

Nipponbare (*Oryza sativa ssp japonica*) was used as the WT in this study. Nipponbare young seedlings were water cultured in a growth chamber (14 h light (28 ± 2 °C)/10 h dark (25 ± 2 °C)) to fourteen-day-old, and then treated with 100 µM ABA (Sigma, St Louise, MO, USA) for the indicated time spans. To avoid the interference of circadian rhythm on the gene transcription, the samples were treated at different starting time to ensure the sample collection done at the same time point. Collected seedling samples were immediately used for RNA extraction.

The *SAPK10* over-expression vector was constructed by PCR amplification of the full CDS and ligation into vector pU1301, in which *SAPK10* was driven by a maize ubiquitin promoter. To make the RNAi construct, around 400 bp long specific fragment of *SAPK10* CDS was amplified and cloned into pANDA as previously reported [25]. Primer sequences could be found in Appendix A. Genetic transformations of the constructs into Nipponbare were done by following [26]. All the plants were grown in the experimental field (natural LD in summer: 14 h light/10 h dark) or greenhouse (natural SD in winter: 10 h light/14 h dark) of china national rice research institute in Hangzhou, China.

### 3.2. Real-Time PCR

The total RNAs from rice young seedlings were extracted using the Trizol (Invitrogen, Carlsbad, CA, USA) reagent according to the manufacturer’s instructions. Firstly, the total RNAs were treated with DNase I (Takara, Dalian, China) in order to remove genomic DNA contamination. RNAs (2 µg) were reversely transcribed into cDNA by using M-MLV reverse transcriptase (Takara, Dalian, China). Real-Time PCR system (10 µL) was constituted with 5 µL of THUNDERBIRD SYBR^®^ qPCR Mix (Toyobo, Shanghai, China) 2 µL of reverse-transcribed product above and 0.2 µM of each primer. *Actin* (*LOC_Os03g61970*) was used as the internal control. The primers of qRT-PCR are listed in Appendix A.

### 3.3. Yeast-two-Hybrid Assay

The ProQuest two-hybrid system (Invitrogen, Carlsbad, CA, USA) was used for the yeast-two -hybrid assays. The full CDS of *bZIP10*, *bZIP11*, *bZIP35*, *bZIP52*, *bZIP55*, *bZIP7*, *bZIP75*, *bZIP76*, *bZIP77* and *bZIP83* were cloned into the pDEST32 vector to generate bait vectors by using the Gateway Technology (Invitrogen, Carlsbad, CA, USA). The CDS of *SAPK1-10* were cloned into pDEST22 vector using the same approach above. Both bait and prey vector plasmids were co-transformed into the yeast strain Y2HGold according to the manufacturer’s instructions (Clontech, Dalian, China). The co-transformed yeast cells were grown on SD/-Trp-Leu solid medium, and further screened on SD/-Trp-Leu-Ade-His solid medium with X- α-Gal to verify the interaction based on the visualization and color of yeast colonies. Each combination was repeated three times, and only those were positive in all three replicates were reported as positive.

### 3.4. In Vitro Pull-Down Assay

The CDS of *OsbZIP77* and *SAPKs* were constructed into pET-28a and pGEX-4T-1 to fuse with 6xHIS and GST tags, respectively. Fused proteins were expressed in the *Escherichia coli* BL21 (DE3) strain and purified by using the Glutathione Sepharose 4 Fast Flow (GE Health care, Chicago, IL, USA) or Ni-NTA Super flow resin (Yisheng biotechnology, Shanghai, China) accordingly. Pull-down was conducted as described by [27]. Briefly, 500 μg of each recombinant protein were mixed with 50 μL Glutathione High Capacity magnetic Agarose Beads in 600 μL pull-down buffer (50 mM Tris-HCl, pH 7.5, 5% glycerol, 1 mM EDTA, 1 mM DTT, 1 mM PMSF, 0.01% Nonidet P-40, 150 mM KCl) for 2 h at 4 ℃. The bounded proteins were pulled down by a Magnetic separation Rack, washed with pull-down buffer twice, and finally eluted with 10 µL 6×SDS and 50 µL PBS. The pulled down proteins were separated on 10% SDS-PAGE gel and were analyzed by western-blot with the primary antibodiesanti-HIS (Cat: CW0083, CWBIO, Beijing, China) and anti-GST (Cat: CW0085, CWBIO, Beijing, China) (1: 5000 dilution), and the secondary antibody anti-mouse conjugated the HRP (1:2000 dilution), respectively.

### 3.5. Kinase Assay

HIS-bZIP77 was co-expressed with GST-SAPK10 and GST tag in *Escherichia coli* BL21 (DE3), respectively, and purified using the Ni-NTA Super flow resin (Yisheng biotechnology, Shanghai, China). The purified proteins (100 ng) were incubated with CIAP (Takara, Dalian, China) at 37 °C for 30 min, and then subjected to western blot analysis of phosphorylated proteins using phos-tag^TM^ BTL-Bound streptavidin-conjugated HRP (PBSH) purchased from Wako (http://www.Phos-tag.com). The PBSH was prepared as below: the solution containing 1xTBST, 1 mM phos-tag^TM^ BTL, 10 mM Zn (NO_3_)_2_, 30 µg Streptavidin-conjugated horseradish peroxidase was set at room temperature for 30 min. Then, the mixed solutions were centrifuged at 14,000× *g* for 20 min with filter device (NanasepTM30k, pall life science, New York, NY, USA) to remove excess phos-tag^TM^ BTL, and then the remaining liquid (<10 µL) was diluted to 30 mL with 1×TBST and placed at 4 °C for later use.

## 4. Conclusions

Overall, by systematically screening the protein-protein interactions of ten SAPKs and nine ABA-inducible bZIPs, we identified 14 positive interaction pairs. bZIP77/OsFD1, which is a key rice flowering regulator, physically bind to SAPK9 and SAPK10, and could be phosphorylated by SAPK10. Similar to the condition in bZIP77/OsFD1, over-expression of SAPK10 resulted in earlier flowering of the plants at least partially through regulating the FAC-MADS15 pathway. our results provided an overall view of the SAPK-bZIP interactions, and shed novel lights on the mechanisms of ABA-regulated rice flowering.

## Figures and Tables

**Figure 1 ijms-20-01427-f001:**
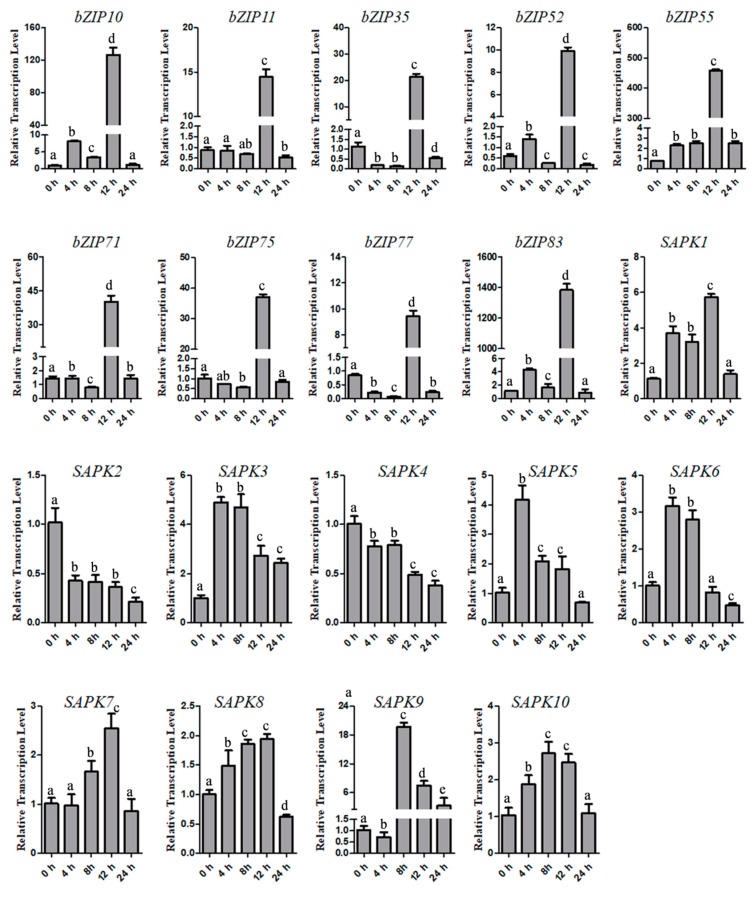
qRT-PCR analysis of *bZIP10*, *bZIP11*, *bZIP35*, *bZIP52*, *bZIP55*, *bZIP71*, *bZIP75*, *bZIP77*, *bZIP83* and *SAPK1-10* in response to 100 µM ABA treatment at 0, 4, 8, 12 and 24 h. Each sample performed with three biological replicates (*n* = 3). Different characters indicate statistical differences at *p* < 0.05 by *students*’ *t*-test. Primer sequences could be found on Appendix A.

**Figure 2 ijms-20-01427-f002:**
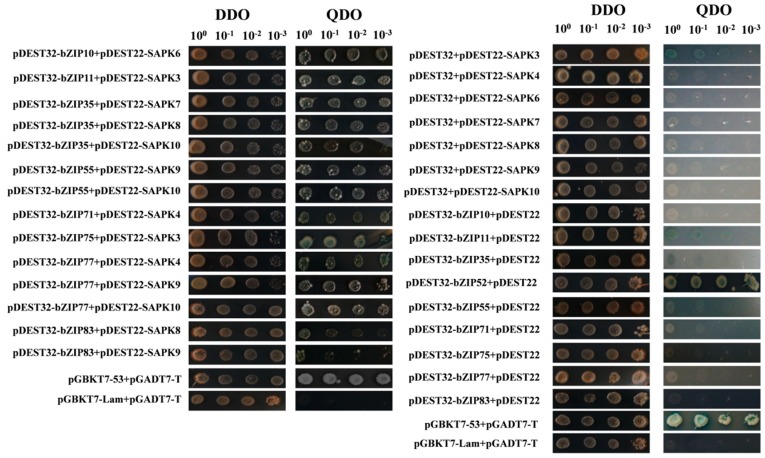
Y2H assays of interactions between SAPKs and ABA-inducible bZIPs. pGBKT7-Lam + pGADT7-T and pGBKT7-53 + pGADT7-T were used as negative and positive controls as instructed by the Kit. DDO: double drop out medium (SD/-Trp-Leu); QDO: quarter drop out medium (SD/-Trp-Leu-Ade-His/ + X-α-Gal). 10^0^ indicate the using of 5 μL amount of yeast cells in OD_600_ = 0.1 for striking. The numbers above the pictures indicate the dilution folds of yeast for tittering on the medium. Each combination was repeated three times, and only those were positive in all three replicates were reported as positive.

**Figure 3 ijms-20-01427-f003:**
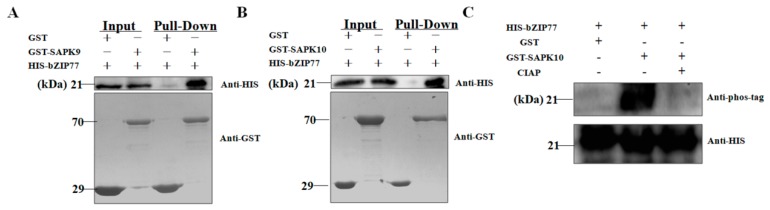
bZIP77 physically interacts with SAPK9 and SAPK10, and phosphorylated by SAPK10. In vitro pull-down assays of GST-SAPK9 and HIS-bZIP77 (**A**), and GST-SAPK10and HIS-bZIP77 (**B**). (**C**) Kinase assay of SAPK10 on bZIP77.

**Figure 4 ijms-20-01427-f004:**
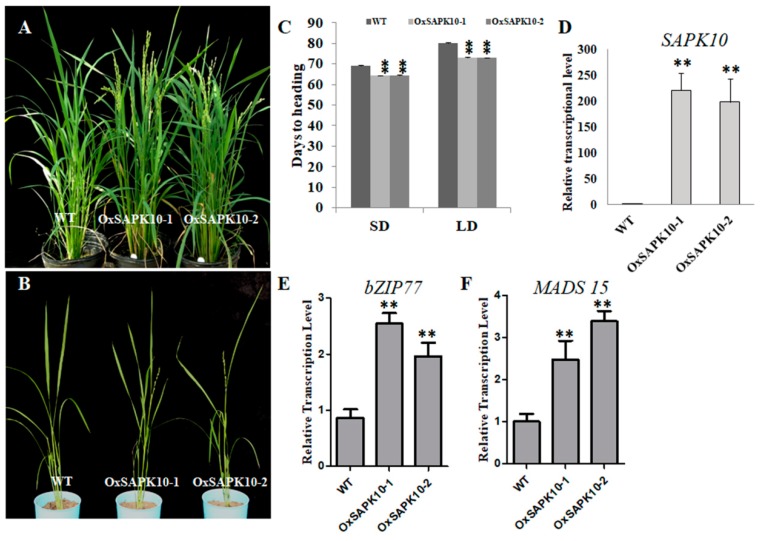
Phenotypical and transcriptional analysis of OxSAPK10 lines. (**A**,**B**) Earlier flowering phenotypes of OxSAPK10 flowering lines under LD (**A**) and SD (**B**) conditions. (**C**) Quantifications of the heading date of OxSAPK10 lines under LD and SD conditions (*n* = 5). (**D**–**F**) Transcriptional levels of *SAPK10*, *bZIP77* and *MADS15* in the WT and OxSAPK10 lines. Data is presented as mean ± SD, ** indicates *p* < 0.01 by *students*’ *t*-test. Each sample performed with three technical replicates (*n* = 3).

**Table 1 ijms-20-01427-t001:** Summary of the interactions between SAPKs and ABA-inducible bZIPs.

	SAPK1	SAPK2	SAPK3	SAPK4	SAPK5	SAPK6	SAPK7	SAPK8	SAPK9	SAPK10
**bZIP10**						**Y**				
**bZIP11**			**Y**							
**bZIP35**							**Y**	**Y**		**Y**
**bZIP52**	**Auto**	**Auto**	**Auto**	**Auto**	**Auto**	**Auto**	**Auto**	**Auto**	**Auto**	**Auto**
**bZIP55**									**Y**	**Y**
**bZIP71**				**Y**						
**bZIP75**			**Y**							
**bZIP77**				**Y**					**Y**	**Y**
**bZIP83**								**Y**	**Y**	

Y: positive interaction; Auto indicates self-activation of the protein in Y2H system. Locus ID of each protein is provided in Appendix B.

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
