# Peer review of "Protein Interactomic Analysis of SAPKs and ABA-Inducible bZIPs Revealed Key Roles of SAPK10 in Rice Flowering"

_ijms, 2019, doi:10.3390/ijms20061427_

Round 1
Reviewer 1 Report
- Please explicit gene names in Abstract and in main text.
- Lines 47-48: “SnRK2-meidated phophorylation on ABI5”, it should be …-mediated phosphorylation of ABI5.
- Line 48: explicit the activity of FLC.
- Line 64-67: why authors wrote SnRK2s if they analyzed the respective rice family SAPK?
- Lines 74-76: is there any reference?
- Line 83: I would delete “and may be functionally involved in ABA signaling” as no evidence are reported.
- Line 83: no statistical analysis is reported for qRT-PCR experiments.
- Line 84-87: “Robust transcriptional induction was only found in SAPK9, which showed over 20 folds up-regulation at 8 HAT, while the other genes only showed significant, but moderate changes of gene expression, which is consistent to the previous studies” As in this case there is no reported correlation between high levels of mRNA and activity of protein, I would only comment the obtained results without stress the sentence with “while the other genes only showed significant”.
- Lines 76-78: “Some ABA-inducible bZIPs, such as bZIP23, bZIP46 and TRAB1/bZIP66 were not included in the assay, as they have been systematically screened with all the SAPKs in previous studies” it not clear what authors would state.
-Lines 87-88: “This result suggested that ABA may impose effects on the expression of SAPKs in the transcriptional or post-translational levels” why authors gave this conclusion? Please comment on it.
-Paragraph “2.2. Interactomic analysis of SAPKs and ABA-inducible bZIPs in yeast” need an introducing sentence explaining why authors performed a Y2H assay.
- No statistical analysis was provided for Y2H experiment presented in Figure2.
-Lines 97-98: “Among the 14 positive interactions, bZIP35 and bZIP77 were interactive with three SAPKs, suggesting their extensive roles in ABA signaling” why authors state that bZIP35 and bZIP77 have “extensive roles in ABA signaling” if they interact with 3 SAPK instead of interacting with 2 like bZIP55 and bZIP83?.
-Lines 111-112: “we did in vitro GST pull-down assay for the combination of SAPK9-bZIP77 and SAPK10-bZIP77” Knowing that authors focus on genes involved in the regulation of flowering time, from the Y2H assay emerged the putative interaction not only of SAPK9-bZIP77 and SAPK10-bZIP77, but also of SAPK4-bZIP77, SAPK7-bZIP35, SAPK8-bZIP35 and SAPK10-bZIP35. Authors should verify with the pull-down assay also these results to highlight the goodness of the Y2H assay.
-Lines 113: “which is a major research topic in our lab” I would say “as we are focusing on molecular mechanism involved in flowering”.
-Lines 113-115: “It has been reported as OsFD1, and could form that FAC (florigen activation complex) with 14-3-3 and Hd3a heterodimer in the nuclear to promote the expression of key flowering regulator OsMADS15” The sentence is not clear, authors need to rephrase the sentence and improve the english. Moreover, it is not clear why authors needs this information. Please comment on it.
-Lines 123-125: “HIS-OsbZIP77 was phosphorylated by GST-SAPK10, but not GST-SAPK9 and GST tag, and the CIAP treatment could remove the phosphorylated band, suggesting that the phosphorylation is highly specific” The sentence is not clear, please rephrase it. Moreover, CIAP treatment remove 5' phosphates. I never heard about this kind of kinase assay, could authors comment on it and provide a reference if it was already used in other works? Maybe authors could also refer to 10.1007/978-1-4939-3115-6_15 and 10.1038/ncomms8981 and provide new experiments confirming the link between the phosphorylation status and the SAPK10-bZIP77 interaction. In addition, authors wrote “HIS-OsbZIP77 was phosphorylated by GST-SAPK10, but not GST-SAPK9 and GST tag”, but I did not found any kinase assay experiment on the GST-SAPK9-HIS-OsbZIP77 in figure. Please comment on it.
-Figure S1B: please explicit the statistical analysis, not only p-value and test used, but also n and number of replicates. Please add “not significant” where differences are not statistically significant.
-Lines 138-141: “Two representative lines OxSAPK10-1 and OxSAPK10-2 were chosen for the experiment due to their highly over-expressed levels when compared with the WT (Figure 4D). As expected, the lines displayed around 14 days and 7 days of earlier heading than the WT in LD and SD, respectively (Figure 4A-C)”. It has been showed that SAPK8, SAPK9 and SAPK10 overexpressing lines exhibited delayed germination and reduced seedling growth (10.1038/ncomms8981). Do the authors have any idea on how their results could fit on the published data?
-Line 140: Why “As expected”? Please comment on it
-Lines 150-151, Figure 6: “Morphologies of OxSAPK10 flowering lines under LD (A) and SD (B) conditions”. Morphology and phenotype are not described or analyzed in this work.
-Lines 145-146: “Meanwhile, the up-regulation of bZIP77 in OxSAPK10 lines hinted comprehensive effects of SAPK10 on bZIP77 in both transcriptional and post-translational-levels” This is not clear. Moreover, why at transcriptional and post-translational levels? On which experiments authors refer to?
-No statistical analysis/comprehensive statistical analysis was provided for the experiments reported. Please indicate the statistical details of each experiments like tests used, p-value, value of n and how many replicates were performed.
Author Response
Response to reviewer 1
-Please explicit gene names in Abstract and in main text.
Response: done
-Lines 47-48: “SnRK2-meidated phophorylation on ABI5”, it should be …-mediated phosphorylation of ABI5. Line 48: explicit the activity of FLC.
Response: corrected
-Line 64-67: why authors wrote SnRK2s if they analyzed the respective rice family SAPK?
Response: According to the previous publication, SnRK2s in rice were named as SAPKs, we changed the name back to SAPKs in the resubmission.
-Lines 74-76: is there any reference?
Response: This database has no references to acknowledge; therefore we only provided the web link for the readers.
-Line 83: I would delete “and may be functionally involved in ABA signaling” as no evidence are reported.
Response: Deleted.
-Line 83: no statistical analysis is reported for qRT-PCR experiments.
Response: Statistics were included.
-Line 84-87: “Robust transcriptional induction was only found in SAPK9, which showed over 20 folds up-regulation at 8 HAT, while the other genes only showed significant, but moderate changes of gene expression, which is consistent to the previous studies” As in this case there is no reported correlation between high levels of mRNA and activity of protein, I would only comment the obtained results without stress the sentence with “while the other genes only showed significant”.
Response: We deleted the part as you suggested.
-Lines 76-78: “Some ABA-inducible bZIPs, such as bZIP23, bZIP46 and TRAB1/bZIP66 were not included in the assay, as they have been systematically screened with all the SAPKs in previous studies” it not clear what authors would state.
Response: We attempted to provide readers the logic that why we did not do the full list of ABA-responsive bZIPs. Since their protein-protein interaction relationships have been elucidated, we didn’t have to repeat that again.
-Lines 87-88: “This result suggested that ABA may impose effects on the expression of SAPKs in the transcriptional or post-translational levels” why authors gave this conclusion? Please comment on it.
Response: We deleted this to avoid confusion.
-Paragraph “2.2. Interactomic analysis of SAPKs and ABA-inducible bZIPs in yeast” need an introducing sentence explaining why authors performed a Y2H assay.
Response: We added the reasons for using Y2H in this study.
-No statistical analysis was provided for Y2H experiment presented in Figure2.
Response: Y2H results are determined by the visualization of the blue colonies on SD/-Trp-Leu-Ade-His/+X-α-Gal medium. We repeated the Y2H for each three times, and only those were positive in all three replicates were reported as positive. To the best of our knowledge, we did not see any statistics on Y2H results in the reported cases.
-Lines 97-98: “Among the 14 positive interactions, bZIP35 and bZIP77 were interactive with three SAPKs, suggesting their extensive roles in ABA signaling” why authors state that bZIP35 and bZIP77 have “extensive roles in ABA signaling” if they interact with 3 SAPK instead of interacting with 2 like bZIP55 and bZIP83?.
Response: We were trying to convey the information that the more interactive SAPKs they have, the more versatile roles they may have in ABA signaling, assuming that different SAPKs may have different functions in this process.
-Lines 111-112: “we did in vitro GST pull-down assay for the combination of SAPK9-bZIP77 and SAPK10-bZIP77” Knowing that authors focus on genes involved in the regulation of flowering time, from the Y2H assay emerged the putative interaction not only of SAPK9-bZIP77 and SAPK10-bZIP77, but also of SAPK4-bZIP77, SAPK7-bZIP35, SAPK8-bZIP35 and SAPK10-bZIP35. Authors should verify with the pull-down assay also these results to highlight the goodness of the Y2H assay.
Response: We totally agree that more pull-down verification assays will make the Y2H results more solid. However, Y2H has been considered as a promising technique in protein-protein interaction detection, particularly under our scenario with high stringency selections (HIS drop out and LacZ selection) and three experimental repeats. We believe our Y2H result is reliable, which is also actually verified by the two pull-down assays. In the current study, the pull-down assays of SAPK9-bZIP77 and SAPK10-bZIP77 were used to verify the Y2H results as well as to explore the possibilities of ABA signaling on rice flowering, while the other suggested combinations have neither ABA-inducible SAPKs nor flowering-related bZIPs. More verification experiments using pull-down or other methods will be conducted in our future works.
-Lines 113: “which is a major research topic in our lab” I would say “as we are focusing on molecular mechanism involved in flowering”.
Response: We deleted this part.
-Lines 113-115: “It has been reported as OsFD1, and could form that FAC (florigen activation complex) with 14-3-3 and Hd3a heterodimer in the nuclear to promote the expression of key flowering regulator OsMADS15” The sentence is not clear, authors need to rephrase the sentence and improve the english. Moreover, it is not clear why authors needs this information. Please comment on it.
Response: This part has been rephrased.
-Lines 123-125: “HIS-OsbZIP77 was phosphorylated by GST-SAPK10, but not GST-SAPK9 and GST tag, and the CIAP treatment could remove the phosphorylated band, suggesting that the phosphorylation is highly specific” The sentence is not clear, please rephrase it. Moreover, CIAP treatment remove 5' phosphates. I never heard about this kind of kinase assay, could authors comment on it and provide a reference if it was already used in other works? Maybe authors could also refer to 10.1007/978-1-4939-3115-6_15 and 10.1038/ncomms8981 and provide new experiments confirming the link between the phosphorylation status and the SAPK10-bZIP77 interaction. In addition, authors wrote “HIS-OsbZIP77 was phosphorylated by GST-SAPK10, but not GST-SAPK9 and GST tag”, but I did not found any kinase assay experiment on the GST-SAPK9-HIS-OsbZIP77 in figure. Please comment on it.
Response: we rephrased the sentence and included the kinase assay picture of SAPK9 as supplemental figure S1. In addition to DNA and RNA, CIAP could also remove phosphorylations on serine, threonine and tyrosine residues of proteins, which has been a common practice in biochemistry and proteomic researches. Several companies like Sigma provided standard online protocols and references for this issue (https://www.sigmaaldrich.com/china-mainland/zh/technical-documents/protocols/biology/roche/alkaline-phosphatase-protocol.html). (J. BIOL. CHEM. 256, 757-760. Cell Biol. 114,735-43.)
-Figure S1B: please explicit the statistical analysis, not only p-value and test used, but also n and number of replicates. Please add “not significant” where differences are not statistically significant.
Response: Corrected
-Lines 138-141: “Two representative lines OxSAPK10-1 and OxSAPK10-2 were chosen for the experiment due to their highly over-expressed levels when compared with the WT (Figure 4D). As expected, the lines displayed around 14 days and 7 days of earlier heading than the WT in LD and SD, respectively (Figure 4A-C)”. It has been showed that SAPK8, SAPK9 and SAPK10 overexpressing lines exhibited delayed germination and reduced seedling growth (10.1038/ncomms8981). Do the authors have any idea on how their results could fit on the published data?
Response: As our focus is on rice flowering, we did not pay attention to these reported traits.
-Line 140: Why “As expected”? Please comment on it
Response: It is deleted.
-Lines 150-151, Figure 6: “Morphologies of OxSAPK10 flowering lines under LD (A) and SD (B) conditions”. Morphology and phenotype are not described or analyzed in this work.
Response: This sentence was rephrased as “The early flowering phenotype of OxSAPK10 lines under LD (A) and SD (B) conditions”
-Lines 145-146: “Meanwhile, the up-regulation of bZIP77 in OxSAPK10 lines hinted comprehensive effects of SAPK10 on bZIP77 in both transcriptional and post-translational-levels” This is not clear. Moreover, why at transcriptional and post-translational levels? On which experiments authors refer to?
Response: We deleted this part.
-No statistical analysis/comprehensive statistical analysis was provided for the experiments reported. Please indicate the statistical details of each experiments like tests used, p-value, value of n and how many replicates were performed.
Response: More details were included. It is a motivation and warm encouragement for the authors to receive such detailed and suggestive comments on our paper. We sincerely appreciate your efforts.
Reviewer 2 Report
Liu et al investigated function of rice SnRK2s by two experiments; searching targets of SnRK2s from bZIP transcription factors that were induced by ABA treatment, and producing SSPK10 overexpression plants to examine the biological role in planta. The finding that the overexpression of SAPK10 resulted in an early heading is interesting, however, their claim that the biological function of SAPK10 remains untouched so far (line 131-132) is not true. SAPK10 was overexpressed in rice and the experiments indicated that SAPK10 is involved in root hair development (Front. Plant Sci., 28 June 2017 | https://doi.org/10.3389/fpls.2017.01121). Did the plants produced in this study showed similar phenotypes to those in the previous study? The authors should not ignore such important previous works. The finding of the target of SAPK10, bZIP77, is also an intriguing possibility to reveal the function of SAPK10, although the authors showed no evidences that bZIP77 is involved in rice flowering (line 112-113). I cannot find any paper about it. If the evidences had been indicated, the findings in this study had great significance. The conclusion described in Abstract, “SAPK10 could phosphorylate bZIP77, and resulted in earlier flowering time when it was over-expressed”, can make misunderstanding. The authors indicated no evidences that, in planta, bZIP77 was phosphorylated by overexpression of SAPK10 and the phosphorylation induced early heading. One can propose a possible model but should declared that it is a model. Similar claim that can make misunderstanding is found in the line 67-69.
Other points
1. In Figure 2, the legend for the numbers above the picture is insufficient. Is it OD600? Also, “10-3” under DDO in the left panel is incomplete.
2. The sentence in line 113-115 is unclear.
3. In any experiments, sample numbers should be indicated (Figure 1, and Figure 6C, E, and F). For qPCR analyses, “biological” triplicate, at least, is required but not “technical”, as shown in Materials and methods (line 178-179). If the experiments were performed for one biological sample in three technical repeats, two additional samples should be investigated.
4. In figure 6F, MASD15 -> MADS15
5. In line 186 X-a-Gal -> “a” should be represented in the font “symbol”.
Author Response
Liu et al investigated function of rice SnRK2s by two experiments; searching targets of SnRK2s from bZIP transcription factors that were induced by ABA treatment, and producing SSPK10 overexpression plants to examine the biological role in planta. The finding that the overexpression of SAPK10 resulted in an early heading is interesting, however, their claim that the biological function of SAPK10 remains untouched so far (line 131-132) is not true. SAPK10 was overexpressed in rice and the experiments indicated that SAPK10 is involved in root hair development (Front. Plant Sci., 28 June 2017 | https://doi.org/10.3389/fpls.2017.01121). Did the plants produced in this study showed similar phenotypes to those in the previous study? The authors should not ignore such important previous works.
Response: Thank you for this reminder. We realized that two previous works investigated the function of SAPK10 in seed germination, post-germination seedling growth and root hair development. However, as our focus is on rice flowering, we did not pay attention to the germination rates and hair roots. Sorry that we can’t provide more information.
The finding of the target of SAPK10, bZIP77, is also an intriguing possibility to reveal the function of SAPK10, although the authors showed no evidences that bZIP77 is involved in rice flowering (line 112-113). I cannot find any paper about it. If the evidences had been indicated, the findings in this study had great significance.
Response: bZIP77/FD1 has been reported as flowering activation complex component. Knock-down of it by RNAi resulted in later flowering, while over-expression of it led to earlier flowering. Only the phosphorylated FD1 could interact with other components to form the complex (doi:10.1038/nature10272, figure 4e and 4f). Therefore, figuring out the upstream kinase of FD1 will be crucial to elucidate the mechanism, and SAPK10 would be one of the very promising candidates, considering the revealed kinase-substrate relationship in this case. We included more discussions regarding its function in flowering and activation by phosphorylation.
The conclusion described in Abstract, “SAPK10 could phosphorylate bZIP77, and resulted in earlier flowering time when it was over-expressed”, can make misunderstanding. The authors indicated no evidences that, in planta, bZIP77 was phosphorylated by overexpression of SAPK10 and the phosphorylation induced early heading. One can propose a possible model but should declared that it is a model. Similar claim that can make misunderstanding is found in the line 67-69.
Response: We corrected the description.
1. In Figure 2, the legend for the numbers above the picture is insufficient. Is it OD600? Also, “10-3” under DDO in the left panel is incomplete.
Response: Corrected. 100 indicate the using of 5 μL amount of yeast cells in OD600=0.1 for striking.
2. The sentence in line 113-115 is unclear.
Response: We rephrase this sentence for a better logic.
3. In any experiments, sample numbers should be indicated (Figure 1, and Figure 6C, E, and F). For qPCR analyses, “biological” triplicate, at least, is required but not “technical”, as shown in Materials and methods (line 178-179). If the experiments were performed for one biological sample in three technical repeats, two additional samples should be investigated.
Response: Sorry, this is a typo. We actually used three biological replicates for the ABA-responsive profile in Figure 1. As for the qPCR analysis in figure 6, we only used three technical replicates for each line, while the data of each line (OxSAPK10-1 and -2) could be used as biological repeats.
4. In figure 6F, MASD15 -> MADS15
Response: Corrected.
5. In line 186 X-a-Gal -> “a” should be represented in the font “symbol”.
Response: Corrected. The authors sincerely appreciate your time and efforts for this paper.
Round 2
Reviewer 1 Report
Please in order to clarify the applied statistics, authors should explicit what letters “a”, “b”, “c”, “d” and “e” in figure 1 state for.
Statistical analysis in figure 1 and figure S2 is still incomplete. Authors should perform at least 3 biological replicates to validate the significance of the experiments, not only three technical replicates as indicated in the relative figure legend. Moreover, the n is still missing.
“biological repeats” should be biological replicates
The rationale why using Y2H is still missing. Please add a sentence explaining why authors would investigate the protein interaction between SAPK and bZIP
Please change “…in our future work” as others could also do it
For the Y2H assay, please indicate “Each combination was repeated three times, and only those were positive in all three replicates were reported as positive” also in the figure legend.
Please be consistent in figure numbering, i.e. Figure 6 should be Figure 4.
Lines 97-98: “Among the 14 positive interactions, bZIP35 and bZIP77 were interactive with three SAPKs, suggesting their extensive roles in ABA signaling” why authors state that bZIP35 and bZIP77 have “extensive roles in ABA signaling” if they interact with 3 SAPK instead of interacting with 2 like bZIP55 and bZIP83?.
Response: We were trying to convey the information that the more interactive SAPKs they have, the more versatile roles they may have in ABA signaling, assuming that different SAPKs may have different functions in this process.
Now lines 99-103: “Among the 14 positive interactions, bZIP35 and bZIP77 were interactive with three SAPKs, suggesting their extensive roles in ABA signaling. In terms of SAPKs, we also found SAPK9 and SAPK10 had more interactive bZIPs than other family members. This observation is consistent with the reported roles of SAPK9 and SAPK10 in ABA signaling [13, 20]”. I do not agree with authors. The information authors would emphasize is cryptic in the text.
Several typos are still present in the text. Moreover, really deep english editing should be done.
Author Response
Response to reviewer 1
Please in order to clarify the applied statistics, authors should explicit what letters “a”, “b”, “c”, “d” and “e” in figure 1 state for.
Response: We actually have made it clear that “Different characters indicate statistical differences at P<0.05” in the legend of figure 1.
Statistical analysis in figure 1 and figure S2 is still incomplete. Authors should perform at least 3 biological replicates to validate the significance of the experiments, not only three technical replicates as indicated in the relative figure legend. Moreover, the n is still missing.
Response: In the last version, we have provided indications that figure 1 was actually done with “THREE BIOLOGICAL repeats” (see figure legend 1), while figure S2 was done with two lines, and three technical replicates for each line. We considered that the two lines are biological replicates for each other. “n=3” is included in the legend of figure 1 as you suggested.
“biological repeats” should be biological replicates
Response: Corrected.
The rationale why using Y2H is still missing. Please add a sentence explaining why authors would investigate the protein interaction between SAPK and bZIP
Response: in the last version, we provided a citation to show the advantages of Y2H, which well explained the rationale for using this technique. As we described in the introduction section, the major theme of this short communication paper is to systematically investigate the interactions between SAPKs and ABA-inducible bZIPs, given that ABA signaling are transmitted through the interaction and phosphorylation of SAPK-ABFs, among which bZIP is the most well-known group so far.
Please change “…in our future work” as others could also do it
Response: Corrected.
For the Y2H assay, please indicate “Each combination was repeated three times, and only those were positive in all three replicates were reported as positive” also in the figure legend.
Response: Done.
Please be consistent in figure numbering, i.e. Figure 6 should be Figure 4.
Response: Corrected. Sorry for this typo.
-Lines 97-98: “Among the 14 positive interactions, bZIP35 and bZIP77 were interactive with three SAPKs, suggesting their extensive roles in ABA signaling” why authors state that bZIP35 and bZIP77 have “extensive roles in ABA signaling” if they interact with 3 SAPK instead of interacting with 2 like bZIP55 and bZIP83?.
Response: We were trying to convey the information that the more interactive SAPKs they have, the more versatile roles they may have in ABA signaling, assuming that different SAPKs may have different functions in this process.
Now lines 99-103: “Among the 14 positive interactions, bZIP35 and bZIP77 were interactive with three SAPKs, suggesting their extensive roles in ABA signaling. In terms of SAPKs, we also found SAPK9 and SAPK10 had more interactive bZIPs than other family members. This observation is consistent with the reported roles of SAPK9 and SAPK10 in ABA signaling [13, 20]”. I do not agree with authors. The information authors would emphasize is cryptic in the text.
Response: We delete the sentences to avoid confusions. Thank you for the remind.
Several typos are still present in the text. Moreover, really deep english editing should be done.
Response: We apologize for the mistakes due to our careless. The language is polished by an English native speaker; we believe the readability is much improved. Thank you again for your efforts concerning the improvement of our manuscript.
Reviewer 2 Report
Liu et al re-wrote the manuscript, however, unfortunately, the revise was incomplete and less careful.
1. Line 25, “Over-expression” and “when it was over-expressed” are redundant.
2. As I pointed out, the sentence in line 69-71 “We identify bZIP77/OsFD1 interaction with SAPK10 and SAPK9 in vitro by Pull‐Down and further find that bZIP77/OsFD1 is phosphorylated by SAPK10 in vitro kinase assays to promote heading time under SDs and LDs.” is difficult to understand and can make misunderstanding. The authors demonstrated that bZIP77 can be phosphorylated by SAPK10, which was just an in vitro assay, but did not demonstrate in vivo phosphorylation nor a direct connection of SAPK10 with bZIP77-involved flowering regulation.
3. The authors likely changed the sentences in Line123-125, but the sentences make no sense. It should be edited. The important description of the result of the SAPK10 dependent phosphorylation has been omitted. The authors should check the sentences carefully and English editing services may be required for the revised sentences.
4. The authors added description about the role of OsFD1 (bZIP77) by citing the paper by Tsuji et al (2013), but the paper did not describe the study of OsFD1, but mainly OsFD2. The authors should cite more appropriate papers.
5. The authors should cite a paper about the OsFD1 over-expressed plants for the sentence in line 150-151.
Author Response
Response to reviewer 2
Line 25, “Over-expression” and “when it was over-expressed” are redundant.
Response: We delete this sentence.
2. As I pointed out, the sentence in line 69-71 “We identify bZIP77/OsFD1 interaction with SAPK10 and SAPK9 in vitro by Pull‐Down and further find that bZIP77/OsFD1 is phosphorylated by SAPK10 in vitro kinase assays to promote heading time under SDs and LDs.” is difficult to understand and can make misunderstanding. The authors demonstrated that bZIP77 can be phosphorylated by SAPK10, which was just an in vitro assay, but did not demonstrate in vivo phosphorylation nor a direct connection of SAPK10 with bZIP77-involved flowering regulation.
Response: In the last revision, we corrected it in section 2.4, but ignored the descriptions here. It is corrected now. We apologize for this careless mistake.
3. The authors likely changed the sentences in Line123-125, but the sentences make no sense. It should be edited. The important description of the result of the SAPK10 dependent phosphorylation has been omitted. The authors should check the sentences carefully and English editing services may be required for the revised sentences.
Response: It is corrected. The manuscript has been edited by a native English speaker, and we believe the readability is much improved.
4. The authors added description about the role of OsFD1 (bZIP77) by citing the paper by Tsuji et al (2013), but the paper did not describe the study of OsFD1, but mainly OsFD2. The authors should cite more appropriate papers.
Response: (Nature, 2011, 476(7360): 332-335) is the right reference, we corrected it.
5. The authors should cite a paper about the OsFD1 over-expressed plants for the sentence in line 150-151.
Response: It is the same reference as you mentioned in the comment above; we provide the citation here now. The authors sincerely appreciate your efforts for this manuscript.
Round 3
Reviewer 1 Report
I really appreciate the effort that authors have made in improving the manuscript. Actually, in the present form the readability has improved and the result section is way more clear.
Some typos are still present:
Figure 3 legend: ...assaysof; bZIP77..
Author Response
Response to reviewer 1
1. Some typos are still present:
Response: We apologize for the mistakes due to our careless. All the typos in manuscript were corrected carefully. The authors sincerely appreciate your efforts for this manuscript again.
2. Figure 3 legend: ...assaysof; bZIP77..
Response: Corrected.